# Absence of Adiponutrin (PNPLA3) and Monoacylglycerol Lipase Synergistically Increases Weight Gain and Aggravates Steatohepatitis in Mice

**DOI:** 10.3390/ijms22042126

**Published:** 2021-02-20

**Authors:** Matteo Tardelli, Francesca V. Bruschi, Claudia D. Fuchs, Thierry Claudel, Nicole Auer, Victoria Kunczer, Onne A. H. O. Ronda, Henkjan J. Verkade, Tatjana Stojakovic, Hubert Scharnagl, Michael Trauner

**Affiliations:** 1Hans Popper Laboratory of Molecular Hepatology, Division of Gastroenterology and Hepatology, Department of Internal Medicine III, Medical University of Vienna, 1090 Vienna, Austria; mat4005@med.cornell.edu (M.T.); francesca_bruschi@yahoo.it (F.V.B.); claudia.fuchs@meduniwien.ac.at (C.D.F.); thierry.claudel@meduniwien.ac.at (T.C.); nicole.auer@meduniwien.ac.at (N.A.); victoria.kunczer@meduniwien.ac.at (V.K.); 2Center for Liver, Digestive and Metabolic Diseases, Departments of Pediatrics, University Medical Center Groningen, 9712 Groningen, The Netherlands; o.a.h.o.ronda@gmail.com (O.A.H.O.R.); h.j.verkade@umcg.nl (H.J.V.); 3Clinical Institute of Medical and Chemical Laboratory Diagnostics, University Hospital Graz, 8036 Graz, Austria; stojakovic@gmx.at; 4Clinical Institute of Medical and Chemical Laboratory Diagnostics, Medical University of Graz, 8036 Graz, Austria; hubert.scharnagl@medunigraz.at

**Keywords:** NAFLD, MGL, PNPLA3, NASH, inflammation

## Abstract

Altered lipid metabolic pathways including hydrolysis of triglycerides are key players in the pathogenesis of nonalcoholic fatty liver disease (NAFLD). Whether adiponutrin (patatin-like phospholipase domain containing protein-3—PNPLA3) and monoacylglycerol lipase (MGL) synergistically contribute to disease progression remains unclear. We generated double knockout (*DKO*) mice lacking both *Mgl* and *Pnpla3*; *DKO* mice were compared to *Mgl^−/−^* after a challenge by high-fat diet (HFD) for 12 weeks to induce steatosis. Serum biochemistry, liver transaminases as well as histology were analyzed. Fatty acid (FA) profiling was assessed in liver and adipose tissue by gas chromatography. Markers of inflammation and lipid metabolism were analyzed. Bone marrow derived macrophages (BMDMs) were isolated and treated with oleic acid. Combined deficiency of *Mgl* and *Pnpla3* resulted in weight gain on a chow diet; when challenged by HFD, *DKO* mice showed increased hepatic FA synthesis and diminished beta-oxidation compared to *Mgl^−/−^.*
*DKO* mice exhibited more pronounced hepatic steatosis with inflammation and recruitment of immune cells to the liver associated with accumulation of saturated FAs. Primary BMDMs isolated from the *DKO* mice showed increased inflammatory activities, which could be reversed by oleic acid supplementation. *Pnpla3* deficiency aggravates the effects of *Mgl* deletion on steatosis and inflammation in the liver under HFD challenge.

## 1. Introduction

Nonalcoholic fatty liver disease (NAFLD) has become the most prevalent liver disease in many countries, affecting approximately 25% of the population worldwide [1]. NAFLD is associated with metabolic syndrome, which includes obesity, insulin resistance, dyslipidemia and hypertension [2,3,4]. A subset of patients with NAFLD develop nonalcoholic steatohepatitis (NASH), with (sub)lethal cell stress and inflammation which drives fibrosis progression to advanced liver disease, including cirrhosis and hepatocellular carcinoma (HCC) [5].

In addition to nutritional and environmental factors [6,7], several genetic variants are associated with NAFLD susceptibility and progression [8]. The genetic polymorphism rs738409 in patatin-like phospholipase domain containing protein-3 (PNPLA3) was shown to confer susceptibility to NAFLD in a genome-wide association scan of nonsynonymous sequence variations [9]. This polymorphism encodes for an isoleucine to methionine substitution at position 148, impairing the enzymatic, putative lipolytic function of adiponutrin [10]. Since its identification, the PNPLA3-I148M human variant has broadly been associated with progression of NAFLD and other liver diseases, including HCC [10,11].

However, how PNPLA3 drives NAFLD pathogenesis remains an area of active investigation [12]. PNPLA3 is expressed in hepatocytes, hepatic stellate cells and adipocytes [13]. In our recent work, we showed that the I148M variant influences hepatic stellate cell activation in vitro, promoting inflammatory and cytokine release and increasing immune cell recruitment [14,15]. Despite the strong association of PNPLA3 with human NAFLD/NASH and other liver disease, *Pnpla3^−/−^* mice showed no phenotype, including differences in TG hydrolysis, energy/glucose/lipid homoeostasis or hepatic steatosis/injury after high-fat diet (HFD) challenge [10]. In another study, loss of *Pnpla3* had no effect on body weight or composition, adipose mass and development, independent of whether the mice were fed regular chow or HFD or had been bred in a leptin deficient (*Lep*^ob/ob^) background [10].

Monoacylglycerol lipase (MGL) as the last enzymatic step responsible for triglyceride (TG) hydrolysis into glycerol and fatty acids (FAs), has been connected to monocyte/macrophage activation in cancer [16,17], fibrosis [18] and cholestatic liver disease [19]. *Mgl^−/−^* animals are protected from steatosis and weight gain under HFD challenge [20,21]. Habib et al. demonstrated that lack of *Mgl* prompted fibrosis regression due to autophagy-mediated anti-inflammatory mechanisms in macrophages using a mouse model lacking *Mgl* in the myeloid lineage (*Mgl^Mye−/−^*) challenged with carbon tetrachloride (CCL4) injection [18]. We recently showed how genetic and pharmacological MGL inhibition could revert cholestatic liver disease, by using the *Mdr2 (Abcb4)* knockout model of sclerosing cholangitis/cholestasis [22] and total body knockout mice for *Mgl* challenged with cholestatic diet (3,5-diethoxycarbonyl-1,4-dihydrocollidine—DDC) [19]. We uncovered that arachidonic acid (AA) accumulated in the intestine and via binding nuclear receptors (NRs), such as peroxisome proliferator activated receptor alpha and gamma (PPAR-α, -γ) and farnesoid X receptor, diminished intestinal inflammation and consequently impacted hepatic bile acid (BA) synthesis via fibroblast growth factor 15 [19]. To the best of our knowledge, no data are available on potential interaction of MGL with PNPLA3, and it is unknown whether a multiple enzymatic disruption along the lipolytic cascade could impact TG lipid partitioning/lipid hydrolysis as well as NR signaling.

Since PNPLA3 is also known to interfere with adipose triglyceride lipase (ATGL) activity by interacting with its cofactor, comparative gene identification-58 (CGI-58) [23], also supposedly participating in MG hydrolysis [10], we aimed to determine the possibility of crosstalk with other key enzymes along the lipid hydrolysis pathway. To address how PNPLA3 affects steatohepatitis, we set out to establish a more extreme NAFLD model than single *Pnpla3^−/−^*—namely, by adding *Mgl* deficiency, which is responsible for the last step in TG hydrolysis. Notably, *Mgl^−/−^* animals are protected against steatosis and weight gain on HFD [20,21]. In the current study, we generated and characterized *Mgl-Pnpla3* whole body double knockout (*DKO*) mice to explore the metabolic adaptations in *Pnpla3* deficiency, which may occur as a functional consequence of aberrant lipid signaling. We found that the concomitant deficiency of *Pnpla3* and *Mgl* causes complex changes in FA metabolism and aggravates steatosis and inflammation.

## 2. Results

### 2.1. Concomitant Absence of Mgl and Pnpla3 Increases Susceptibility to Weight Gain and Hepatic Inflammation

To study whether loss of both *Mgl* and *Pnpla3*, as key intracellular lipases involved in hydrolysis of stored fat, would impact lipid partitioning and regulation of nuclear receptor (NR) signaling, we first characterized a *DKO* mouse model at baseline. Importantly, body weights of *DKO* mice were significantly increased in comparison to WT, *Mgl^−/−^* or *Pnpla3^−/−^* at week 8 of chow diet (Figure 1A). When challenged with HFD for 12 weeks, serum transaminases (ALT, AST) as well as serum TG, were strongly increased in *DKO* mice compared to *Mgl^−/−^* (Figure 1B), whereas serum cholesterol and NEFA remained unchanged (Figure 1C, Appendix A). Further analysis showed increased fat content in the liver (Figure 1D) accompanied by increased liver weight (0.85 g, SD = 0.15 versus 1.3 g SD = 0.22 for *Mgl^−/−^* and *DKO*, respectively) and TG content (Figure 1E). Surprisingly, inflammation was strongly increased in *DKO* animals as shown in the IHC staining for Mac-2 (Figure 1D) together with gene expression of inflammatory markers such as *Tnfa*, *Ccl2*, and *iNos* (Figure 1F) and protein expression of NFkb and IKba (Figure 1G—with respective quantification in the bar graph). Expression of genes involved in FA storage and synthesis was increased (Pparg2—gene expression and protein, *Dgat1* and *Fasn*), whereas genes involved in beta-oxidation were downregulated (*Ppara, Aox*), at least in part consistent with accumulating lipids in the liver (Figure 1H). Taken together, our data showed that *DKO* mice have increased lipid storage and inflammation in liver, likely due to increased lipid deposition, diminished FA oxidation, and increased serum triglycerides.

### 2.2. DKO Mice Show a Depleted Pool of Unsaturated FAs in the Liver with Enriched Saturated Species in the Liver and AT

In order to explain the serum TG increase, we analyzed the expressions of key genes involved in hepatic very low-density lipoprotein (VLDL) synthesis and clearance. Expressions of *Mtp* and *ApoB100* genes were higher in *DKO* mice than in *Mgl^−/−^,* whereas *ApoC3*, the LPL inhibitor, was unchanged (Figure 2A). In line with this, the relative (%) of non-HDL-cholesterol fraction in serum confirmed the gene expression data by showing an increased abundance of non-HDL lipoproteins (Figure 2B). Next, FA species were analyzed in liver and AT homogenates by gas chromatography. In the liver, the relative contribution of saturated FAs (SFAs) to the total amount of FAs present in the liver was higher in *DKO* compared to *Mgl^−/−^,* whereas monounsaturated fatty acids (MUFAs) and polyunsaturated fatty acids (PUFAs) (in trend) were lower (Figure 2C); in particular, 22:6w3, 18:2w6, and 20:3 w6 were significantly decreased (Figure 2D). In AT, SFA concentrations increased whereas MUFA remained unchanged and PUFA decreased in *DKO* (Figure 2E). Specifically, we found that SFAs such as myristic and palmitic acid strongly increased, while 18:3w3, 18:2w6 decreased, and 20:4w6 increased (Figure 2E).

### 2.3. Absence of Mgl and Pnpla3 Increases White Adipose Tissue Lipid Storage and Inflammation

We further investigated the liver to body weight (BW) ratio that was increased trend-wise, whereas gonadal adipose tissue (GWAT) to BW ratio was significantly increased in *DKO* compared to *Mgl^−/−^* (Figure 3A). GWAT analysis showed increased expression of FA synthesis genes as shown by *Srebp1c, Dgat1*, and *Pparg2* mRNA expressions and uptake by *Lpl* (Figure 3B), while *Fasn, adiponectin*, and *leptin* were not statistically increased (Figure 3B and Appendix A). In line with this, inflammatory genes such as *Tnfa, iNos,* and *Il1b* were strongly increased in AT from *DKO* animals (Figure 3B); this observation was strengthened by Mac-2 staining and quantification of crown-like structures (CLSs) (Figure 3C,D). Histological analysis of other adipose depots such as the brown adipose tissue (BAT) did not show significant differences (Figure 3E), with unchanged UCP1 protein expression, but increased *Ucp2, D2,* and *Ppara* genes (Figure 3F).

### 2.4. Primary BMDMs from DKO Mice Are Proinflammatory and Their Phenotypes Can Be Partially Reversed with Oleic Acid Treatment

We aimed to explore whether this profound inflammatory phenotype in the liver and AT was a cause or rather a consequence of increased TG storage. We hypothesized that the inflammatory accrual could originate from recruited immune cells and we therefore isolated bone marrow derived macrophages (BMDMs) as a model for further analysis. Thus, to uncover potential direct effects of FAs on monocyte/macrophage polarization, we investigated BMDM treatments in vitro. Strikingly, the combined knockout of *Mgl* and *Pnpla3* affected the expression of inflammatory genes such as *Il1b, Tnfa*, and *NFkb*, also upregulating cholesterol synthesis pathways via *Srebp2, Hmgcr*, and *Ldlr* (Figure 4A). The proinflammatory prostaglandin E2 was found to be more prevalent in the medium of *DKO* BMDMs (Figure 4B). When this same medium was utilized on U937 NFkb-GFP, their response, reflecting NFkb activation, was much more enhanced by conditioned medium derived from *DKO* (Figure 4C). Furthermore, the presence of exogenous LPS during polarization altered expression of inflammatory markers such as *Il1b* and *Tnfa* (Figure 4D). When BMDMs from *DKO* mice were treated with oleic acid (OA), expression of inflammatory markers decreased, suggesting that inflammation could be reversed with MUFA supplementation (Figure 4E). Collectively these data suggest that the enrichment of different species of FAs drives the inflammatory phenotype in the liver and AT in *DKO* mice (Figure 4F).

## 3. Discussion

We discovered that *Pnpla3* is involved in inflammation and lipid accumulation of different species of FAs in an NAFLD mouse model. The most striking phenotype in the *DKO* mice was the rapid onset of inflammation and steatosis on HFD challenge. This contrasted to the minimal inflammation, diminished body weight and steatosis seen in *Mgl^−/−^* animals on the same diet [20,21], which is usually not expected to trigger steatohepatitis [8,21] in contrast to other established models such as methionine-choline deficient diet [24].

Importantly, gene expression analysis indicates that the human PNPLA3 is expressed in several tissues, but its highest level in the liver, which suggests it has a key role in this organ [12]. Instead, *Pnpla3* expression in mice shows different patterns, being more expressed in adipose depots than in the liver [25]. Notably, important discrepancies have been reported among different cellular populations within the liver, between mice and humans [26]. Due to its close homology with ATGL (approximately 70%), its subcellular localization and strong regulation by nutritional and metabolic stimuli [7] suggest considering PNPLA3 as a key protein involved in the control of lipid homeostasis in metabolic organs, such as liver and AT. Despite the experimental observations that have been collected from its first discovery in 2001 [10], the enzymatic role of PNPLA3 still remains unclear and opposite functions have been reported. On the one hand, PNPLA3 exhibits hydrolase activity towards triglycerides and directly interacts with CGI-58 resulting in ATGL inhibition on lipid droplets [27,28]. On the other hand, acyltransferase activity towards polyunsaturated fatty acids in phospholipids was shown in other works [29,30], suggesting a role in lipid catabolism. The human recombinant PNPLA3 catalyzes the hydrolysis of TGs, DGs and MGs, with higher preference for TGs [29,30], while both human and murine PNPLA3 have been described to function as acyl-CoA-dependent lysophosphatidic acid acyltransferase, thus favoring the synthesis of phosphatidic acid towards TGs [31]. In contrast to in these in vitro findings, in vivo knockout strategies failed to elucidate the metabolic role of this protein, since mice lacking *Pnpla3* generated by two independent groups displayed no metabolic and phenotypical alterations, even not when challenged with several metabolic insults (i.e., refeeding, high-sucrose diet, diet-induced obesity, methionine-choline deficient diet or leptin deficiency background) [32,33]. Of note, only hepatic mRNA *Pnpla3* was induced after introduction of high sucrose lipogenic diet, in a similar pattern to PNPLA3, but with no further consequence on ATGL expression [33]. In addition, no compensatory effect on other TG lipases or lipogenic enzymes have been reported in *Pnpla3* knockout mice, suggesting a nonexclusive role of this protein in TGs metabolism in vivo, at least in mice [33].

In the present study, we challenged these findings and generated a *DKO* strategy in which *Pnpla3* deletion would be accompanied by *Mgl* ablation to investigate their roles in NAFLD. Since the human PNPLA3 protein supposedly takes part in MG hydrolysis, we deleted *Mgl* as the rate-limiting enzyme in MG metabolism [34], rendering one of the substrates more available along the TG hydrolysis pathway.

The lack of both lipases profoundly exacerbated hepatic steatosis and, importantly, increased inflammation in the liver and AT. Similarly to *Pnpla3-148M* knockin mice, high-sucrose diet increased hepatic fat content [35], whilst in another study steatohepatitis was even ameliorated by knocking down *Pnpla3* in hepatocytes [36]. These in vivo observations implicate hepatocyte-expressed PNPLA3-148M as a loss of function responsible for the pathogenesis of NAFLD. In our model, the absence of both lipases depleted the pool of MUFAs and PUFAs in the liver relative to increasing SFAs in the liver and AT. However, this finding contrasts with a human study in which PNPLA3-I148M patients had increased hepatic retention of PUFA [37]. The striking difference in FA species between AT and liver may be responsible for the phenotypic changes in FA metabolism and inflammation we describe, as FAs are potent ligand of NRs. We speculate that the observed phenotype of *DKO* mice is likely a consequence of *Pnpla3* deletion and depends on the activation of alternative signaling pathways triggered by accumulation of certain saturated FAs species in the concomitant absence of *Pnpla3* and *Mgl.* In addition, since PNPLA3 was observed to have maximal hydrolytic activity against TG, DG, and MG with a strong preference for oleic acid as the acyl moiety [10,13], we corroborate that these lipid classes would accumulate in the liver driving macro- or microvesicular steatosis. To verify this hypothesis, attentive lipidomic and proteomic analyses are required to understand the role of protein–protein interactions amongst other lipases such as ATGL and its cofactor CGI-58 in response to this disruption.

Our in vitro finding that inflammation decreases with OA treatment in BMDMs is supported by the observation that the latter are key contributors to hepatic macrophage accumulation in experimental steatohepatitis [38]. Accordingly, another study found out that by using bone marrow transplantation, BMDMs were the predominant population contributing to the enhanced macrophage infiltration in steatohepatitis [39]. Moreover, depletion of hepatic BMDMs by liposomal clodronate during liver injury attenuated fibrosing steatohepatitis and inflammation in methionine-choline deficient diet (MCD) and HFD models [39]. We thus associate the changes in FA species in liver resulting by parallel deletion of *Pnpla3* and *Mgl* with the inflammatory phenotype in *DKO* mice.

In conclusion, these and other mechanistic differences between the PNPLA3 variant and its contribution to NAFLD progression remain to be further elucidated. Moreover, the very involvement of immune cells in the liver following invalidation of lipases suggests that further studies in humans regarding NAFLD/NASH and other liver diseases are warranted.

## 4. Materials and Methods

### 4.1. Animal Experiments

Experiments were performed in 3-month-old male *Mgl^−/−^* and *Mgl-Pnpla3* double knockout (*DKO*) mice (C57BL/6J background, *n* = 8 per group unless stated otherwise) weighing 25–30 g. Animals were housed in a 12 h light/dark house facility with ad libitum consumption of water and food. They received either high-fat diet (60% fat) or standard chow (A04 from SAFE diets, Augy, France) for 12 weeks. The experimental protocols were approved by the local Animal Care and Use Committee (BMWF.66.009/0117-II/3b/2013).

### 4.2. Serum Biochemistry

Routine serum biochemical analyses were performed as previously described [19].

### 4.3. Liver Histology, Immunohistochemistry, and CLS Quantification

Livers were fixed in 4% neutral buffered formaldehyde solution for 24 h and embedded in paraffin. Sections of 3 μm thickness were stained with hematoxylin and eosin (H&E) or Mac-2, as described elsewhere [21]. Crown-like structures (CLSs) were quantified with Image J 1.8 [40].

### 4.4. Western Blotting

Tissues were collected in RIPA buffer and protein concentration measured using 660 nm protein assay kit. Proteins extracts were loaded on an SDS-PAGE using 10% polyacrylamide gels to investigate transporter expression of P65, IKBa, PPARg, UCP1 (all 1:500, Santa Cruz Biotechnology, Dallas, TX, USA) and Calnexin (1:1000) [19].

### 4.5. Gas Chromatography

Liver samples (approximately 10 mg) were subjected to FA isolation and derivatization. All TGs, phospholipids, and cholesterol esters were split up into free fatty acids (FFAs) and derivatized by the methyl donor (acidic methanol). A known quantity of C17 was added to each sample as internal standard. From the internal standard, we calculated the values of each fatty acid species. After methylation, the sample was concentrated in hexane and injected into the gas chromatography system. The adipose tissue was homogenized in chloroform using a tight Dounce pestle and tube. AT (5 mg equivalent) and an internal standard (C17) were methylated in acid methanol, and FFAs were extracted twice using hexane, concentrated to 200 μL under a stream of N2, and analyzed on GC-FID. FFAs were quantified and expressed as umol/mg [19].

### 4.6. Gene Expression

Gene expression levels were determined by quantitative real-time PCR (Thermo Fisher Scientific, Waltham, MA, USA) using RNA extracted from livers, GWAT, and brown adipose tissue (BAT) [19].

### 4.7. Hepatic Lipid Concentrations

Reagent kits (Wako Diagnostics, Richmond, VA, USA) were used to determine hepatic concentrations of triglycerides, cholesterol, and free fatty acids, as previously performed [21].

### 4.8. BMDMs Isolation LPS and OA Treatments

Bone marrow cell suspensions were isolated by flushing femurs and tibias of 8- to 12-week mice with complete RPMI1640 (+10% FBS, +1% Pen/Strep—all Thermo Fisher Scientific, Waltham, MA, USA). Aggregates were dislodged by gentle pipetting, and debris was removed by passaging the suspension through a 70 µm nylon web [40]. Cells were supplemented with 20 ng/mL M-CSF (Thermo Fisher Scientific, Waltham, MA, USA) and cultured in a humidified incubator at 37°C and 5% CO2. They were harvested at indicated time points by gentle pipetting and repeated washing of the wells with phosphate buffered saline (PBS), 0.5% bovine serum albumin (BSA), and 2 mM EDTA to detach adherent cells [40] (all Thermo Fisher Scientific, Waltham, MA, USA). Cells were either treated with 20 ng/mL of LPS or 100 mg/mL OA (Sigma-Aldrich, St. Louis, MO, USA). The concentrations of PGE2 in media were determined by quantitative ELISA according to the manufacturer’s directions (Enzo Biosciences, Lausanne, Switzerland).

### 4.9. U937-NF-κB GFP

The human monocytic cell line U937 (ATCC^®^ CRL-1593.2™, Manassas, VI, USA) was stably transfected with a GFP promoter under the control of NF-κB. U937-NF-κB was cultured in RPMI-1640 medium supplemented with 10% fetal bovine serum (FBS), 2 mM glutamine, 100 IU/mL penicillin, 100 μg/mL streptomycin, and 1 mM Na-pyruvate. Cells were maintained in a humidified atmosphere (5% CO_2_) at 37 °C. All materials were obtained from Sigma-Aldrich (St. Louis, MO, USA).

### 4.10. Statistical Methods

Differences between groups were analyzed using a two-tailed unpaired Student’s *t*-test (Prism v6.0h, GraphPad, La Jolla, CA, USA). Data are represented as mean ± S.D. for multiple experiments.

## Figures and Tables

**Figure 1 ijms-22-02126-f001:**
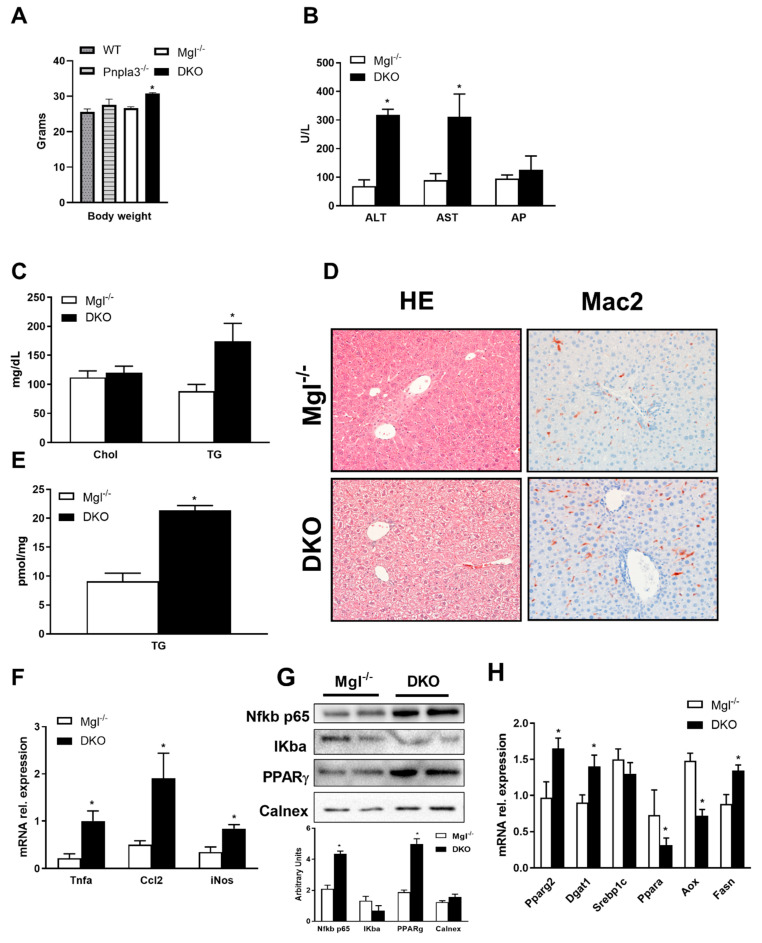
Absence of both *monoacylglycerol lipase* (*Mgl*) and *patatin-like phospholipase domain containing protein-3* (*Pnpla3*) increase susceptibility to weight gain and hepatic inflammation. (**A**) Body weight measurement revealed increased weight gain in *double knockout* (*DKO*) compared to other genotypes on a chow diet. (**B**) Serum biochemistry reflects increased levels of transaminases (ALT, AST—but unchanged AP) as well as (**C**) increased TG levels and unchanged cholesterol in *DKO* on a high-fat diet (HFD). (**D**) Representative hematoxylin and eosin (H&E) images (10× magnification) with markedly increased steatosis and inflammation (Mac-2) in *DKO* versus *Mgl^−/−^*. (**E**) Increased liver TGs in *DKO* on a HFD. (**F**) Hepatic gene expression of proinflammatory markers *Tnfα, Ccl2*, and *iNos* increased together with (**G**) protein expressions and respective quantifications of NFKb p65 and PPARg whilst IKba were decreased. (**H**) Hepatic gene expressions of *Pparg2, Dgat1,* and *Fasn* increased, whereas beta-oxidation decreased as demonstrated by *Ppara* and *Aox* genes. Results are expressed as mean ± S.D.; * *p* < 0.05 for *Mgl^−/−^* versus *DKO* mice.

**Figure 2 ijms-22-02126-f002:**
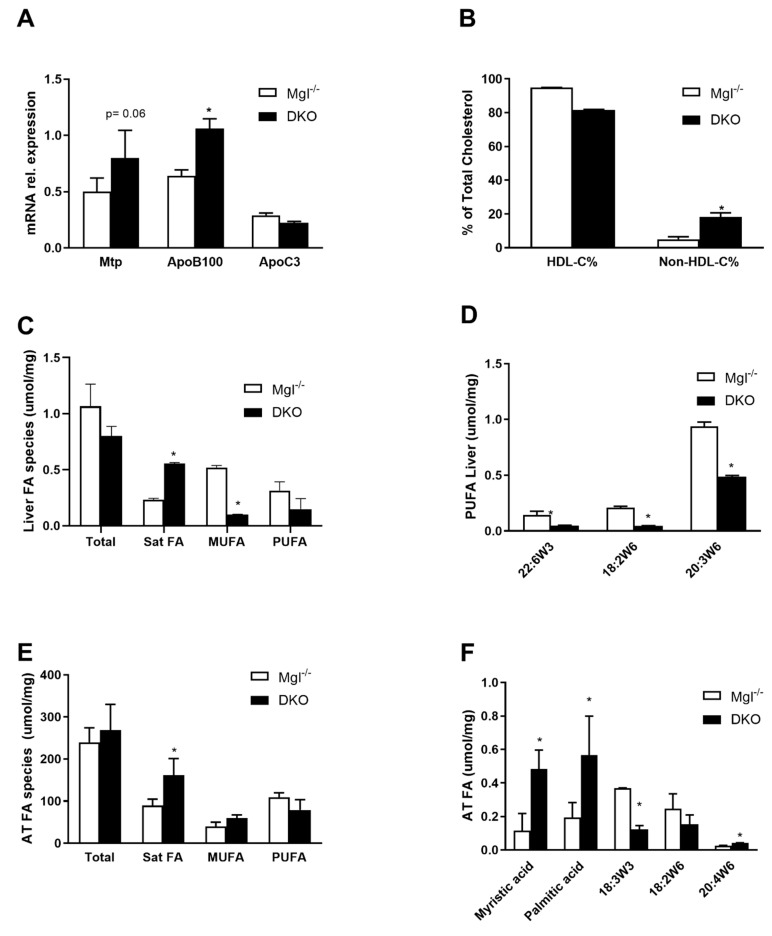
*DKO* shows enriched fatty acid (FA) species in liver and AT. (**A**) Gene expression of genes involved in very low-density lipoprotein (VLDL) production was increased for *Mtp* and *ApoB100,* whereas the *ApoC3* gene for HDL was unchanged. (**B**) HDL-C% and respective non-HDL-C% were measured in serum. (**C**) Gas chromatography quantification of hepatic FA evidenced accumulation of saturated FAs with a decrease in monounsaturated fatty acids (MUFAs) and polyunsaturated fatty acids (PUFAs) in *DKO* (**D**) of 18:2w6 and 20:3w6 diminished in *DKO*. (**E**) In the AT, saturated fatty acids (SFAs) accumulated with (**F**) increased mystiric acid, palmitic acid, and decreased 18:2w3/6. Results are expressed as mean ± S.D.; * *p* < 0.05 for *Mgl^−/−^* versus *DKO* mice.

**Figure 3 ijms-22-02126-f003:**
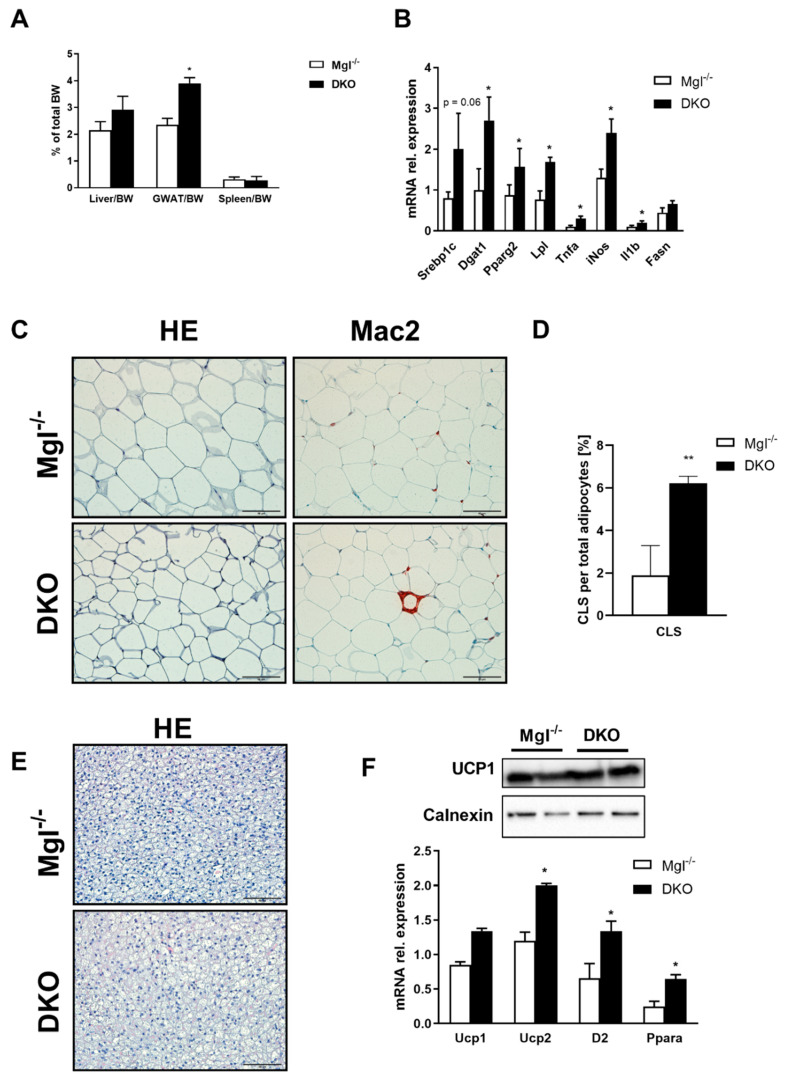
*DKO* mice show increased FA synthesis and inflammation in AT. (**A**) Liver/body weight (BW) (trend-wise) and gonadal adipose tissue (GWAT)/BW ratio increased in *DKO,* whereas spleen/BW did not change. (**B**) Gene expression of *Srebp1c, Dgat1, Pparg2, Lpl,* and *Fasn*(in trend) increased in *DKO,* whereas inflammatory markers increased for *Tnfa, iNos,* and *Il1b*. (**C**) Representative H&E images (10× magnification) with markedly increased inflammation (Mac-2) in *DKO* versus *Mgl^−/−^.* (**D**) Crown-like structures (CLSs) were highly increased in *DKO,* as quantified by ImageJ. (**E**) Representative H&E images (10× magnification) of brown adipose tissue (BAT) in *DKO*. (**F**) Ucp1 protein expression and gene expression did not change, whereas *Ucp2, D2,* and *Ppara* increased. Results are expressed as mean ± S.D.; ** *p < 0.01,* * *p* < 0.05 for *Mgl^−/−^* versus *DKO* mice.

**Figure 4 ijms-22-02126-f004:**
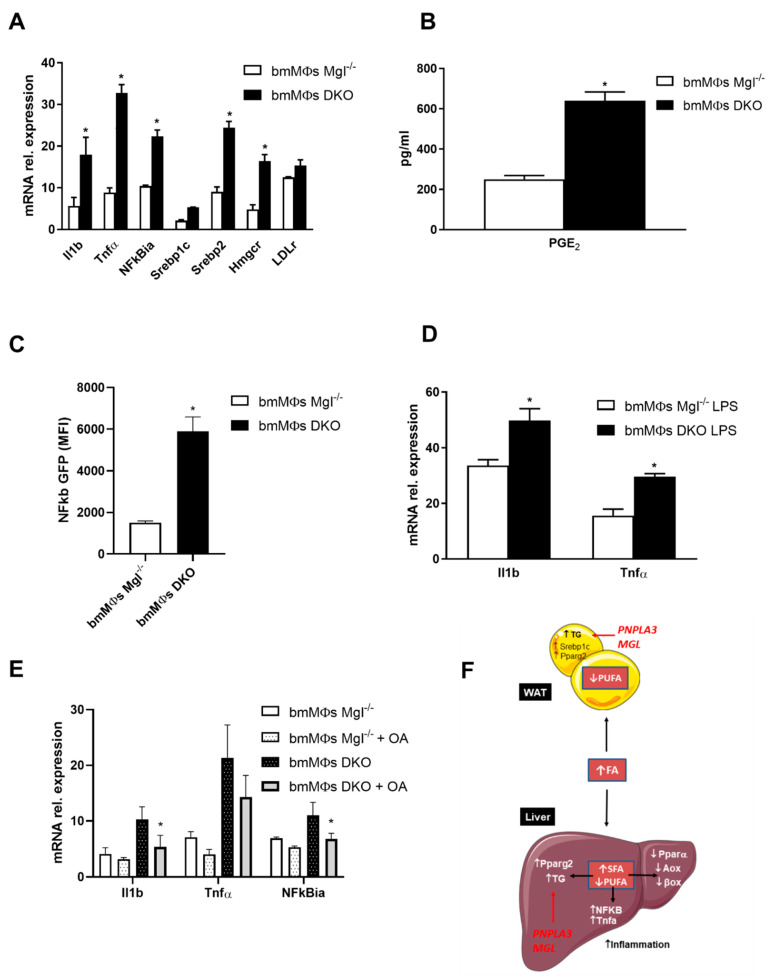
Bone marrow derived macrophages (BMDMs) from *DKO* mice are proinflammatory and their phenotypes can be partially reversed with oleic acid (OA) treatment. (**A**) Gene expression of proinflammatory *Il1b, Tnfa,* and *nfkbia* increased along cholesterol synthesis as shown by *Srebp2, Hmgcr,* and *LDLr*. (**B**) Increased concentration of proinflammatory PGE2 in BMDM medium. (**C**) BMDM conditioned media strongly activated NFKB pathway in U937. (**D**) When treated with LPS, gene expression of *Il1b* and *Tnfa* further increased in BMDMs from *DKO*. (**E**) Treatment with OA diminished gene expressions of *Il1b, Tnfa,* and *Nfkbia* in BMDMs from *DKO*. (**F**) Proposed mechanism of action showing decreased PUFA in the AT driving TG accumulation and inflammation, with concomitant increase in SFA in the liver and TG with inflammation and decreased b-oxidation. Results are expressed as mean ± S.D.; * *p* < 0.05 for BMDMs *Mgl^−/−^* versus BMDMs *DKO*.

## Data Availability

The data presented in this study are available in the present article.

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
