# Peer review of "Absence of Adiponutrin (PNPLA3) and Monoacylglycerol Lipase Synergistically Increases Weight Gain and Aggravates Steatohepatitis in Mice"

_ijms, 2021, doi:10.3390/ijms22042126_

Round 1
Reviewer 1 Report
In the present manuscript, Tardelli et al. investigate the genetic interaction between adiponutrin (PNPLA3) and monoacylglycerol lipase (MGL) using double knock-out mouse model. Their findings show a deterioration of metabolic parameters including increased body weight gain, hepatic steatosis and inflammation of the double knock-out mouse. The authors find in addition that fatty acid oxidation in decreased in the double knock-out model compared to the single MGL knock-out mouse. This particular finding led them to conclude that absence of PNPLA3 aggravates metabolic homeostasis in a MGL deficient model.
The manuscript is elegantly presented and shows clear and convincing findings. The experiments presented are well designed and used the best tools available to answer the questions the authors are asking. Using these mouse models elevates the physiological relevance of the author’s findings. Overall, the manuscript has an excellent quality.
The phenotype is clear and solid, there is not a clear molecular mechanism proven to explain this phenotype however, it is well understood that it is not the scope of the paper. On the other hand, the authors could speculate a little bit in the discussion section about potential mechanisms. This addition would elevate better the scope and future directions of the paper.
Regarding specific experiments, it is clear the body weight of the double ko mice is increased and that this is also associated with increased adipose tissue mass. This is well addressed in figure 3A, but it should be explained which adipose tissue depot was used. In addition, specify in the Y-axis % of total body weight.
It seems like there is an increased fatty acid synthesis in both adipose tissue and liver and that Pparg is involved. Are downstream regulators also affected like AcetyCoA Carboxylase (ACC) by its mRNA levels or phosphorylation status? Or fatty acid synthase?
Particularly for the liver, it seems clear an increased lipid content with the H&E staining in Fig 1D, however, an even better proof would be to do an oil red O staining.
Finally, correct all formatting of gene and protein symbol according to the consensus:
Gene
PROTEIN
Author Response
In the present manuscript, Tardelli et al. investigate the genetic interaction between adiponutrin (PNPLA3) and monoacylglycerol lipase (MGL) using double knock-out mouse model. Their findings show a deterioration of metabolic parameters including increased body weight gain, hepatic steatosis and inflammation of the double knock-out mouse. The authors find in addition that fatty acid oxidation in decreased in the double knock-out model compared to the single MGL knock-out mouse. This particular finding led them to conclude that absence of PNPLA3 aggravates metabolic homeostasis in a MGL deficient model.
The manuscript is elegantly presented and shows clear and convincing findings. The experiments presented are well designed and used the best tools available to answer the questions the authors are asking. Using these mouse models elevates the physiological relevance of the author’s findings. Overall, the manuscript has an excellent quality.
The phenotype is clear and solid, there is not a clear molecular mechanism proven to explain this phenotype however, it is well understood that it is not the scope of the paper. On the other hand, the authors could speculate a little bit in the discussion section about potential mechanisms. This addition would elevate better the scope and future directions of the paper.
We thank the Reviewer for the overall positive feedback, and we appreciate the constructive criticisms to our study. We included an additional section in the discussion part, speculating further about this phenotype (page 10 lines 254-260).
Regarding specific experiments, it is clear the body weight of the double ko mice is increased and that this is also associated with increased adipose tissue mass. This is well addressed in figure 3A, but it should be explained which adipose tissue depot was used. In addition, specify in the Y-axis % of total body weight.
We adapted Figure 3A as suggested, changed the Y-axis and specified that we used GWAT in new figure 3, text (results line 135-136, M&M line 310) and figure legend.
It seems like there is an increased fatty acid synthesis in both adipose tissue and liver and that Pparg is involved. Are downstream regulators also affected like AcetyCoA Carboxylase (ACC) by its mRNA levels or phosphorylation status? Or fatty acid synthase?
We thank the Reviewer for this comment. We measured fatty acid synthase (Fasn) gene in the liver and adipose tissue and included the results in new Figure 1 and Figure 3. Fasn upregulated significantly and only increased in trend in liver and GWAT, respectively. We adapted figure legend and text accordingly.
Particularly for the liver, it seems clear an increased lipid content with the H&E staining in Fig 1D, however, an even better proof would be to do an oil red O staining.
This is an excellent suggestion and would add a further body of evidence to this work, however due to technical difficulties we did not prepare tissues for cryosection when this experiment took place. Thus, we unfortunately cannot stain for ORO as mice are not breeding anymore and must rely on the tissues collected and still available. Nevertheless, we hope that the measurement of TG in the liver after lipid extraction (Figure 1E) are convincing enough and could represent a valid surrogate to answer your question.
Finally, correct all formatting of gene and protein symbol according to the consensus: Gene PROTEIN
Many thanks for noticing this, we adapted the style as suggested and carefully corrected throughout the text.
Reviewer 2 Report
Tardelli et al. describe how PNPLA3 interacts with MGL in controlling hepatic lipolysis. The combined deficiency causes an elevation in hepatic steatosis much greater than that seen with ablation of MGL alone. Overall the effects are both convincingly presented and are interesting, moreover, the authors have previously shown the role of PNPLA3 variants in stellate cell activation in vitro. However, I have some queries below which should be addressed.
Minor comments:
Fig1G quantitation is necessary. Calnexin levels in DKO are elevated
Fig2B Can the authors explain the absence of variation in TC levels? SD for the non-HDL-c for DKO is not even observable.
Standard Gene nomenclature formats should be followed throughout including figures. For E.g. Mouse genes are named Xyz123. For instance, in Fig 3F, I presume that authors are referring to deiodinase, iodothyronine, type II (Dio2) referred to as D2 without any abbreviation. All qPCR primer sequences should be provided as supplemental information.
The scheme in Fig 4F should be updated to show PNPLA3 and MGL at the appropriate steps.
The description of statistical methods should be revised for correctness. What do the authors mean by "with corrections for multiple comparisons"
Author Response
Tardelli et al. describe how PNPLA3 interacts with MGL in controlling hepatic lipolysis. The combined deficiency causes an elevation in hepatic steatosis much greater than that seen with ablation of MGL alone. Overall the effects are both convincingly presented and are interesting, moreover, the authors have previously shown the role of PNPLA3 variants in stellate cell activation in vitro. However, I have some queries below which should be addressed.
Minor comments:
Fig1G quantitation is necessary. Calnexin levels in DKO are elevated
We thank the Reviewer for the overall positive feedback, and we appreciate the constructive criticisms to our study. We included the quantification study run with BioRad software just below the blots: Calnexin is just slightly increased according to the blots quantification.
Fig2B Can the authors explain the absence of variation in TC levels? SD for the non-HDL-c for DKO is not even observable.
The variation of total cholesterol is evident in DKO where especially the non-HDL fraction increases. The HDL fraction was instead unchanged consistently with total plasma cholesterol as shown in Fig. 1C. Thanks for the observation, we noticed that SD disappeared due to a technical glitch on GraphPad, we revised Fig. 2B accordingly.
Standard Gene nomenclature formats should be followed throughout including figures. For E.g. Mouse genes are named Xyz123. For instance, in Fig 3F, I presume that authors are referring to deiodinase, iodothyronine, type II (Dio2) referred to as D2 without any abbreviation. All qPCR primer sequences should be provided as supplemental information.
Thanks for noticing this, we adapted the style as suggested and added a full list of abbreviation at the end of the manuscript so to make this clearer for the reader.
The scheme in Fig 4F should be updated to show PNPLA3 and MGL at the appropriate steps.
We updated Figure 4F and added PNPLA3 and MGL at the appropriate steps, as we believe they strongly affect TG levels (and therefore DNL and beta oxidation) in both AT and liver.
The description of statistical methods should be revised for correctness. What do the authors mean by "with corrections for multiple comparisons".
Many thanks for noticing this mistake, we corrected this in the method sections: t-test was used as statistical method.